# Loss of circSRY reduces γH2AX level in germ cells and impairs mouse spermatogenesis

Yanze Song[1,2,3,6,*], Min Chen[4,*,†] , Yingfan Zhang[5], Jiayi Li[1,2,3,6], Bowen Liu[1,2,3,6] , Na Li[1,2,3], Min Chen[1,2,3,6,†], Miaomiao Qiao[1,2,3,6], Nan Wang[1,2,3,6], Yuanwei Cao[1,2,3,6], Shan Lu[1,2,3,6], Jian Chen[1,2,3,6], Wen Sun[1,3,6] , Fei Gao[1,2,3,6] , Haoyi Wang[1,2,3,6]

**Sry on the Y chromosome is the master switch of sex determination in mammals. It has been well established that Sry encodes a transcription factor that is transiently expressed in somatic cells of the male gonad, leading to the formation of testes. In the testis of adult mice, Sry is expressed as a circular RNA (circRNA) transcript. However, the physiological function of Sry circRNA (circSRY) remains unknown since its discovery in 1993. Here we show that circSRY is mainly expressed in the spermatocytes, but not in mature sperm or somatic cells of the testis. Loss of circSRY led to germ cell apoptosis and the reduction of sperm count in the epididymis. The level of γH2AX was decreased, and failure of XY body formation was noted in circSRY KO germ cells. Further study demonstrated that circSRY directly bound to miR-138-5p in spermatocytes, and in vitro assay suggested that circSRY regulates H2AX mRNA through sponging miR-138-5p. Our study demonstrates that, besides determining sex, Sry also plays an important role in spermatogenesis as a circRNA.**

## Introduction

Long non-coding RNAs are relatively abundant in the mammalian transcriptome (Kapranov et al, 2007) and play important roles in development and reproduction (Taylor et al, 2015). Circular RNA (circRNA) is a unique type of non-coding RNA generated through back-splicing to form a covalently linked loop (Memczak et al, 2013; Qu et al, 2015). Since the first circRNA was discovered in the 1970s (Sanger et al, 1976), very few circRNAs have been identified in the following years. In the last decade, however, the development of RNA sequencing technologies and bioinformatics has greatly facilitated the discovery of circRNAs (Salzman et al, 2012).

Many circRNAs were found in various cell types, regulating various biological processes, such as transcription, alternative splicing, chromatin looping, and post-transcriptional regulation (Hentze & Preiss, 2013; Li et al, 2015; Panda et al, 2016; Conn et al, 2017; Guarnerio et al, 2019; Liu et al, 2020). One of the functional mechanisms of circRNA is that they act as competing endogenous RNAs to sponge miRNAs (Poliseno et al, 2010; Salmena et al, 2011; Hansen et al, 2013; Memczak et al, 2013).

Sry is best known as the sex determination gene on the Y chromosome. In mice, Sry is expressed as a transcription factor from 10.5 to 12.5 dpc in the genital ridge somatic cells, initiating testis development (Koopman et al, 1990). The introduction of Sry locus into the female mouse embryo switches the sex to male, whereas the targeted mutation in male embryos leads to complete male-to-female sex reversal (Koopman et al, 1991; Kato et al, 2013; Wang et al, 2013). Moreover, mutation of SRY causes disorders of sex development in humans (Hawkins, 1993). Recently, a cryptic second exon of mouse Sry hidden in the palindromic sequence was identified, and this two-exon Sry transcript plays a primary role in sex determination (Miyawaki et al, 2020). Curiously, Sry is also expressed in adult mouse testis as a circRNA (circSRY) (Capel et al, 1993; Dubin et al, 1995; Hacker et al, 1995). A linear transcript containing long inverted repeats is transcribed from a distal promoter, followed by a back-splicing event that covalently links an acceptor splice site at the 5′ end to a donor site at a downstream 3′ end (Memczak et al, 2013). Although the presence of circSRY in the testis has long been discovered, the significance of circSRY remains elusive.

In this study, we generated mouse models that did not express circSRY, without interfering with male sex determination. We found that circSRY played an important role in spermatogenesis and further dissected the underlying mechanism. Our findings highlight a unique synergy between Sry's male sex determination role in

[1]State Key Laboratory of Stem Cell and Reproductive Biology, Institute of Zoology, Chinese Academy of Sciences, Beijing, China [2]University of Chinese Academy of Sciences, Beijing, China [3]Institute for Stem Cell and Regeneration, Chinese Academy of Sciences, Beijing, China [4]Guangdong and Shenzhen Key Laboratory of Male Reproductive Medicine and Genetics, Institute of Urology, Peking University Shenzhen Hospital, Shenzhen Peking University-Hong Kong University of Science and Technology Medical Center, Shenzhen, P.R. China [5]The Jackson Laboratory, Bar Harbor, ME, USA [6]Beijing Institute for Stem Cell and Regenerative Medicine, Beijing, China

Correspondence: wanghaoyi@ioz.ac.cn; gaof@ioz.ac.cn; sunwen@ioz.ac.cn
*Yanze Song and Min Chen contributed equally to this work
†The same name for different authors

developing embryo as a protein and its regulatory role in male germ cells as a circRNA.

# Results

### Characterization of circSRY in mouse testis

To identify *Sry* transcripts in adult mouse testis, divergent primers and convergent primers were designed to amplify circSRY or *Sry* linear transcripts, respectively (Fig 1A). Upon RNase R treatment, circSRY was still detectable by RT–PCR, whereas linear RNA was not (Fig 1B). The expression of circSRY was abundant when random hexamer primers were used for reverse transcription,

whereas it was barely detectable using oligo $(dt)_{18}$ primers. In comparison, the expression of *Sry* linear RNA was low using either random hexamer primers or oligo $(dt)_{18}$ primers (Fig 1C). The presence of the head-to-tail splicing site of circSRY was verified by Sanger sequencing (Fig S1A). Furthermore, by separating the cytoplasm and nucleus fractions, we found that circSRY was mainly localized in the cytoplasm (Fig S1B). All these results confirmed previous report that *Sry* transcripts in adult mouse testis are non-polyadenylated circRNAs mainly localized in the cytoplasm (Capel et al, 1993)

Next, we measured the level of *Sry* transcription from day P10 to adulthood. The amount of circSRY increased over time, whereas linear RNA was barely detectable (Fig 1D). Using flow cytometry, we separated different types of cells in the testis of 8-wk-old mice:

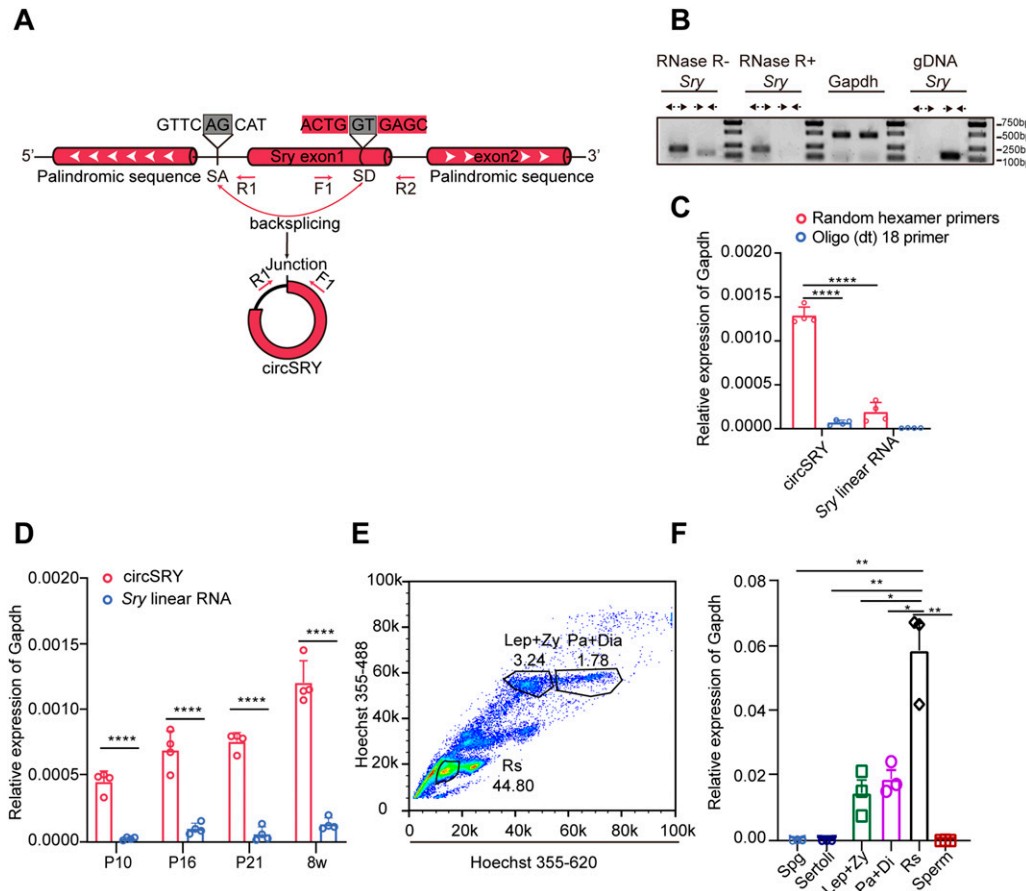

**Figure 1. Identification of circSRY.**
**(A)** Scheme illustrating the generation of circSRY. CircSRY is formed by an incomplete single exon Sry gene through the back-splicing mechanism. Convergent or divergent primers detect circRNA (F1 and R1) or linear RNA (F1 and R2) of Sry. The gray boxes indicate the head-to-tail splicing sequences. **(B)** Production of divergent primers was resistant to RNase R treatment. CircSRY was not amplified using genomic DNA as the template (n = 3 biologically independent experiments). **(C)** Random hexamer primer or Oligo (dt)18 primers were used to analyze the expression of circSRY in 8-wk-old control mice (****P < 0.0001; unpaired t test; n = 4 biologically independent experiments). **(D)** Relative expression of circSRY or linear RNA in testis from 10 d postnatal to adulthood (****P < 0.0001; two-way ANOVA test; n = 4 biologically independent experiments). **(E)** Fluorescence cytometry separated different subtypes of germ cells in adult testis. **(F)** Relative expression of circSRY in six cell groups of (Spg: spermatogonium; Lep + Zy: leptotene stage and zygotene stage; Pa + Di: pachytene stage and diplotene stage; Rs: round spermatids; sperm: mature sperm from epididymis; Sertoli: Sertoli cells; n = 3 biologically independent experiments). The relative expression of circSRY in Spg, Sertoli, and sperm cells was less than 0.0001 (**P < 0.0001,*P < 0.001, unpaired t test; n = 3 biologically independent experiments). Gapdh was used as reference gene. Data information: data are presented as mean ± SEM.
Source data are available for this figure.

leptotene and zygotene spermatocytes (Lep + Zy), pachytene and diplotene (Pa + Di) spermatocytes, and round spermatids (Rs) (Figs 1E and S1C, a–e). Sertoli cells were obtained from P20 testis, spermatogonia (Spg) were obtained from P6 testis through flow cytometry (Fig S1C, f and g and D), and high purity of each cell type was confirmed based on specific markers (Fig S1E). Sperms were obtained from adult mouse epididymis (Fig S1C, h). CircSRY was detected in meiotic and post-meiotic germ cells, expressed at the highest level in Rs. There was no circSRY detected in Spg, sperm, or Sertoli cells (Fig 1F).

## Generation of circSRY KO mouse

To determine whether *Sry* is involved in spermatogenesis, we generated circSRY KO mouse via CRISPR-Cas9 (Fig 2A). We designed a single-guide RNA (sgRNA) targeting the splice acceptor site (SA) upstream of *Sry* coding region. Deletion of the 11 bp region harboring splice acceptor site led to complete loss of circSRY (Fig 2B and C). We euthanized circSRY KO embryos from E11.5 to E14.5 to assess the sex determination and gonad development. No significant morphological difference was detected in control and circSRY

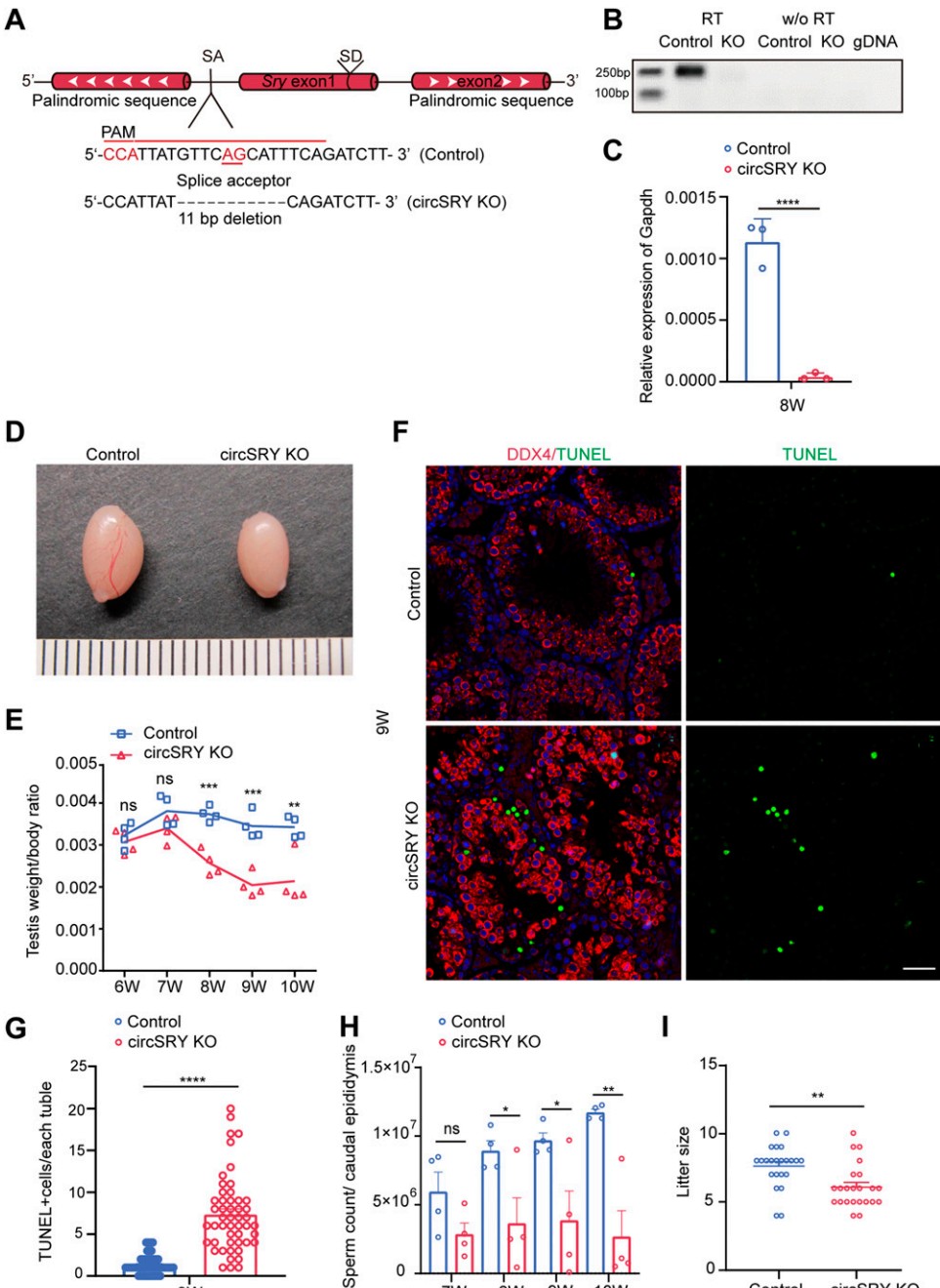

**Figure 2. Loss of circSRY affected fertility of male mice.**
**(A)** Design of deleting circSRY using CRISPR-Cas9; a specific sgRNA was designed to target splicing acceptor site of circSRY. **(B)** RT–PCR results of circSRY from 8-wk-old control or circSRY KO testis (n = 3 biologically independent experiments). **(C)** Expression quantification of circSRY within control or circSRY KO mice testes (****P < 0.0001; unpaired *t* test; n = 3 biologically independent experiments). Gapdh was used as reference gene. **(D)** Representative image of circSRY KO and control (WT) testis of 8-wk-old mice. **(E)** Testis/body ratio of circSRY KO and control mice from 6 to 10 wk of age. *P*-values are presented above the relevant bars (***P < 0.001, **P < 0.01, ns, not significant; unpaired *t* test; n = 4 biologically independent experiments). **(F)** Germ cells were performed with TUNEL assay (green) and labeled with the antibody against DDX4 (red) in the 9-wk-old circSRY KO and mouse testes. (****P < 0.0001; unpaired *t* test). Scale bar indicates 50 μm. **(G)** Number of TUNEL-positive cells within seminiferous tubules in circSRY KO and control 9-wk-old mouse testes (****P < 0.0001, unpaired *t* test; n = 3 biologically independent experiments). **(H)** Sperm count of circSRY KO mice and control mice from 7 to 10 wk of age (*P < 0.05, **P < 0.01, ns, not significant; unpaired *t* test; n = 4 biologically independent experiments). **(I)** Litter size of circSRY KO compared with control mice. Random 8-wk-old males of control and circSRY KO were chosen to breed with 6-wk-old females of C57BL/6 for 4 mo (**P < 0.01, unpaired *t* test; four control or circSRY KO male mice were used for this experiment, each individual point represented one count of litter size). Data information: data are presented as mean ± SEM. Source data are available for this figure.

KO embryo gonads (Fig S2A). The expression of *Sry* transcripts was similar between circSRY and control embryos at E11.5 (Fig S2B) and the circSRY KO embryo gonads at E12.5 and E14.5 developed normally (Fig S2C). We also checked 8-wk-old circSRY KO (XY) mice and found no gross abnormalities of external genitalia (Fig S2D), and they were fertile. In F0 and F1 generations, all the male mice carried Y chromosomes and all the females did not (Fig S2E), indicating that deletion of the 11 bp upstream region of *Sry* did not interfere with sex determination. Female offsprings were normal, and no developmental defects were observed (Fig S2D).

### The fertility of circSRY KO male mice was impaired

To determine the function of circSRY, we assessed the fertility of circSRY KO male mice. The circSRY KO male mice were fertile, but the size of their testes was smaller than that of control (WT C57BL/6) mice at 8 wk (Fig 2D). The testis to body weight ratio of circSRY KO mice was comparable to that of control mice at 6–7 wk, whereas it became significantly lower from 8–10 wk (Fig 2E). The development of germ cells was examined by DDX4 staining. As shown in Fig 2F, the histology of the seminiferous tubules was grossly normal. DDX4-positive germ cells were detected in the testes of both control and circSRY KO mice at 8 wk (Fig 2F). However, there were notably more TUNEL-positive cells in the seminiferous tubules of circSRY KO mice than that of control mice (Fig 2F and G). Furthermore, we found that the total number of sperm in the caudal epididymis of 8–10-wk-old circSRY KO mice was lower than that in control mice (Fig 2H). To evaluate whether this sperm reduction in circSRY KO mice will exacerbate with age, we examined 13-mo-old mice. Albeit the sperm count of circSRY KO mice was significantly lower than that of control group, it was still over two million (Fig S3), suggesting that Spgl stem cells were not severely affected in circSRY KO mice. To test fertility, we crossed 4 circSRY KO males with WT female mice and counted the litter size born within 4 mo. CircSRY KO mice produced a smaller litter size than age-matched control male mice (Fig 2I).

Notably, no difference in SOX9-positive Sertoli cells was detected between control and circSRY KO seminiferous tubules (Fig S3B and C), suggesting that Sertoli cells were not affected. In addition, there is no difference about the percentage of mobile sperm between the control and the circSRY KO mice (Fig S3D), indicating that the loss of circSRY did not affect sperm mobility. Furthermore, we assessed the process of spermiogenesis through periodic acid–Schiff/hematoxylin staining which revealed the maturation of spermatids into spermatozoa. The spermatids differentiated from circSRY KO seminiferous epithelium underwent morphological changes normally and eventually transformed into streamlined spermatozoa (Fig S3E), indicating that circSRY does not participate in the maturation of spermatids into spermatozoa.

To characterize the cell type–specific function of circSRY in spermatogenesis more rigorously, we generated a *Sry* conditional KO mouse model *Sry^{flox}* by inserting two loxP sites flanking *Sry* (Fig S3F). We specifically knocked out *Sry* in germ cells or Sertoli cells by crossing *Sry^{flox}* male mice with *Stra8-Cre* (Sadate-Ngatchou et al, 2008) or *Amh-Cre* (Lecureuil et al, 2002) transgenic mice, respectively (Fig S3F). Compared with control mice, the testis size and the testis to body weight ratio of *Sry^{flox}*; *Stra8-Cre* male was smaller at 2 mo of age, and a lower number of sperm in the caudal

epididymis were detected (Fig 3A and B). Accordingly, decreased DDX4-positive germ cells and an increased number of apoptotic germ cells were observed in *Sry^{flox}*; *Stra8-Cre* mice (Figs 3C and D and S3G). These defects were similar to those observed in circSRY KO mice. By contrast, no defect of germ cell development was observed in *Sry^{flox}*; *Amh-Cre* male mice (Figs 3E–H and S3H), indicating that circSRY is not required for Sertoli cells' function in adult male testis. Taken together, these results showed that specific loss of circSRY in adult male germ cells led to impaired spermatogenesis.

### Loss of circSRY causes reduction of primary spermatocytes during spermatogenesis

To further characterize the defects of spermatogenesis in circSRY KO mice, we examined the expression of meiosis-associated genes by immunofluorescence (IF). PLZF-positive and STRA8-positive germ cells were localized at the periphery of seminiferous tubules, and no difference was detected either for PLZF-positive or STRA8-positive germ cells between control and circSRY KO testes (Fig S3I and J). Notably, the ratio of SYCP3-positive cells versus SOX9-positive cells was reduced within the circSRY KO mice seminiferous tubules (Fig 4A and B). These defects were independent of Sertoli cells, given that the number of SOX9-positive cells was not decreased in circSRY KO mice testes (Fig S3B and C). To assess the progression of spermatogenesis, flow cytometry was used to analyze the proportion of 4N cells in 2-mo-old testes. The proportion of 4N cells was reduced in circSRY KO mice (Fig 4C). In addition, TUNEL and SYCP3 double-positive germ cells were observed in the circSRY KO seminiferous tubules, but not in control mice (Fig 4D). We also assessed primary spermatocytes in the seminiferous tubules of *Sry^{flox}*; *Stra8-Cre* and *Sry^{flox}*; *Amh-Cre* male mice by SOX9 and SYCP3 double staining. The ratio of SYCP3-positive cells to SOX9-positive cells declined in *Sry^{flox}*; *Stra8-Cre* male mice, whereas the number of SOX9-positive cells remained the same with control tubules (Fig 4E–G). Neither the ratio of SYCP3-positive cells to SOX9-positive cells nor the number of SOX9-positive cells was different between *Sry^{flox}*; *Amh-Cre* and control tubules (Fig 4E–G). Taken together, these results indicate that lack of circSRY in germ cells leads to the reduction of spermatocytes.

### Loss of circSRY leads to reduced H2AX mRNA level and increased frequency of X–Y synapsis defect

To elucidate the underlying mechanism of circSRY in regulating spermatogenesis, we collected primary spermatocytes from 8-wk-old control and circSRY KO mice and performed RNA-seq analysis (Fig S4A and B). Gene Ontology (GO) analysis showed DNA repair and meiotic cell cycle were highly enriched terms (GO: 0006281, GO: 0051321, Fig S4C), which was consistent with meiosis-related cell apoptosis observed in circSRY KO mice (Fig 4A–G). We then performed chromosome spreading using spermatocytes from circSRY KO mice with immunostaining of SYCP3 and γH2AX. As shown in Fig 5A, the γH2AX signal in circSRY KO pachynema nucleus was in general weaker than in control nucleus (n = 50) (Fig 5A), which is supported by the decreased H2AX expression in circSRY KO spermatocytes through RNA-seq analysis (logFC = -1.85) (Fig S4D). We next performed qRT–PCR, and a decreased H2AX expression was

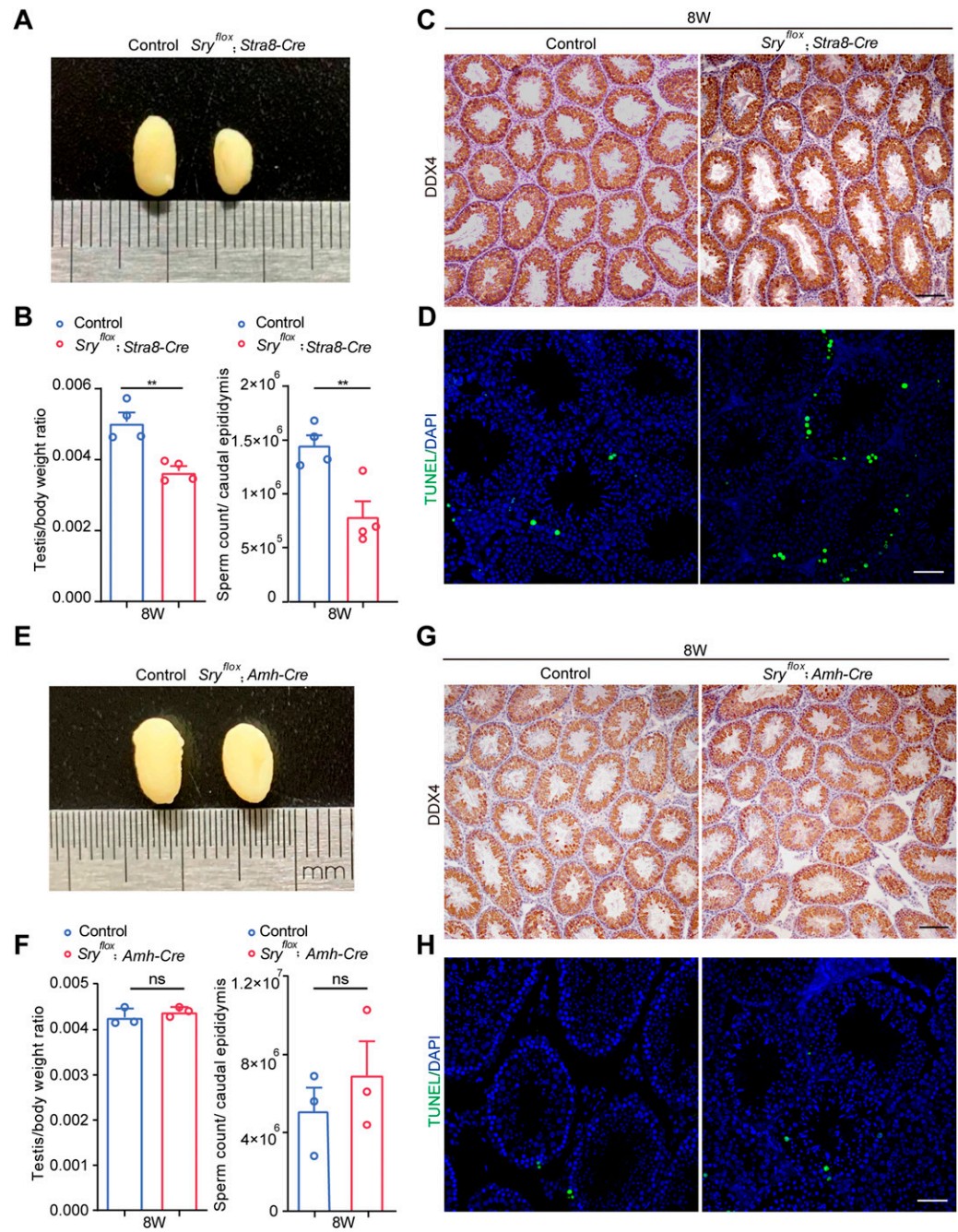

**Figure 3. Conditional knockout of *Sry* resulted in germ cells loss.**
**(A)** Representative image of testis from 8-wk-old mice control (left) and *Sry^flox*; *Stra8-Cre* mice (right) from the same litter. **(B)** Testis/body ratio and sperm count of 8-wk-old control mice compared with *Sry^flox*; *Stra8-Cre* from the same litter. (**P < 0.01, unpaired *t* test; n = 4 biologically independent experiments). **(C)** Germ cells were labeled with the antibody against DDX4 (brown). Loss of epithelium within the seminiferous tubules was observed in 8-wk-old control mice compared with *Sry^flox*; *Stra8-Cre* from the same litter. Scale bar indicates 100 μm. **(D)** TUNEL assay in *Sry^flox*; *Stra8-Cre* and 8-wk-old control mice seminiferous tubules. Scale bar indicates 50 μm. **(E)** Representative image of testis from 8-wk-old mice control (left) compared with *Sry^flox*; *Amh-Cre* mice (right) from the same litter. **(F)** Testis/body ratio and sperm count of 8-wk-old control mice compared with *Sry^flox*; *Amh-Cre* from the same litter (ns, not significant, unpaired *t* test; n = 3 biologically independent experiments). **(G)** Germ cells were labeled with the antibody against DDX4 (brown) in the 8-wk-old control mice compared with *Sry^flox*; *Amh-Cre* from the same litter. Scale bar indicates 100 μm. **(H)** TUNEL assay in *Sry^flox*; *Amh-Cre* and 8-wk-old control mice seminiferous tubules. Scale bar indicates 50 μm. Data information: data are presented as mean ± SEM. Source data are available for this figure.

found in circSRY KO spermatocytes compared with control group (Fig 5B). Western blot analysis further revealed that the γH2AX level was reduced in circSRY KO spermatocytes (Fig 5C).

To further assess the X–Y synapsis during early pachynema, IF staining of SYCP1 and SYCP3 was performed with the chromosome spread experiment. In 15.3% of circSRY KO nuclei (n = 150), we

https://doi.org/10.26508/lsa.202201617   vol 6 | no 2 | e202201617   

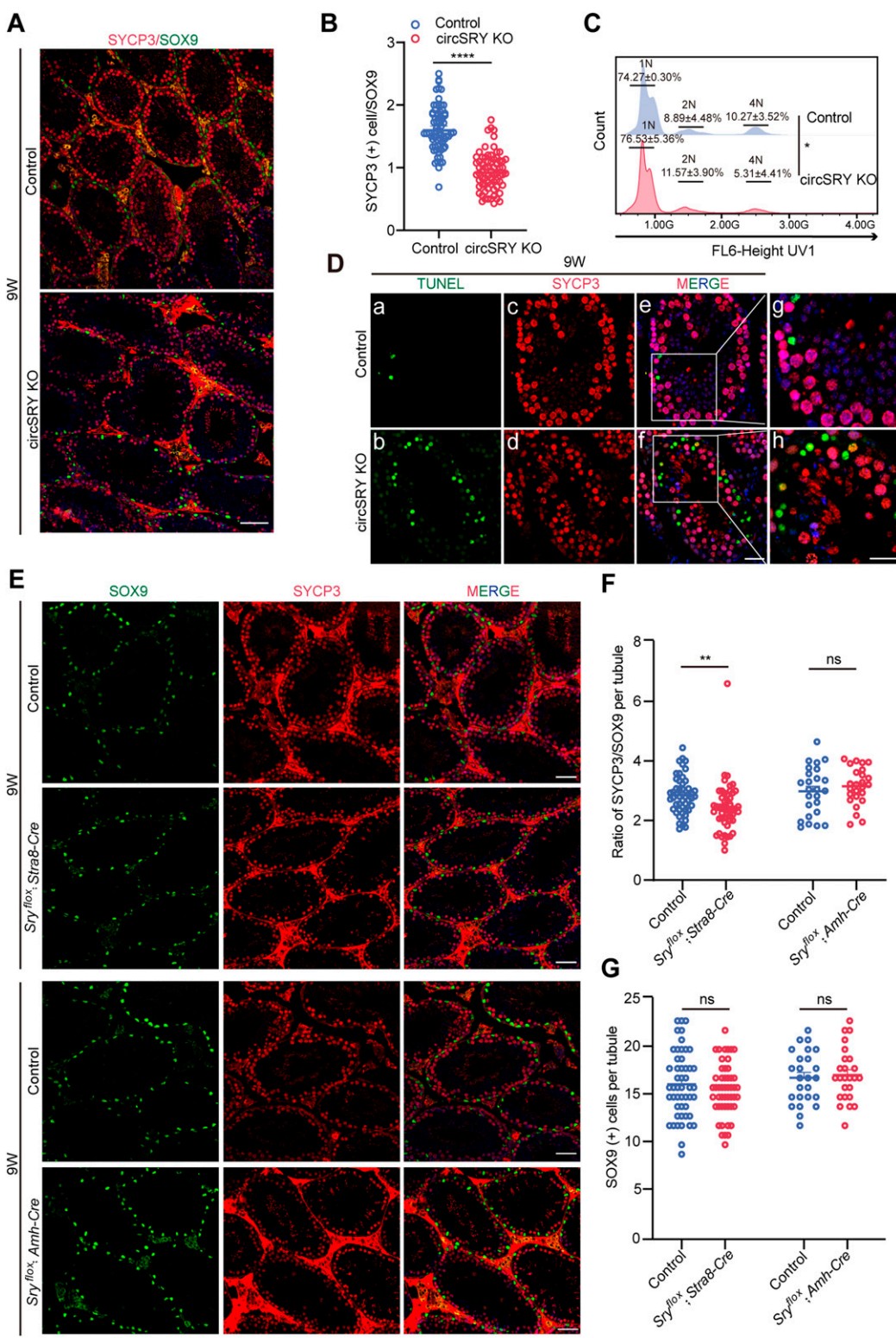

**Figure 4. Loss of circSRY led to the decreased number of primary spermatocytes.**
**(A)** Immunofluorescence staining of SYCP3 (red) and SOX9 (green) in seminiferous tubules of 9-wk-old control and circSRY KO mice. Scale bar indicates 50 $\mu m$.
**(B)** Quantification of SYCP3-positive cells versus SOX9-positive cells in seminiferous tubules from 9-wk-old circSRY KO compared with control mice (****$P < 0.0001$; unpaired $t$ test, total 140 tubules from four biologically independent experiments were counted in this experiment). **(C)** Flow cytometry analysis of the proportion of 4N cells, 1N cells, and 2N cells in 9 wk mice testes between control and circSRY KO mice (*$P < 0.05$; unpaired $t$ test; n = 3 biologically independent experiments). **(D)** (a–h) Representative image of immunofluorescence co-staining of TUNEL signal (green) and SYCP3 (red) in 9-wk-old control and circSRY KO mice testes (a–f, scale bar

observed that X and Y chromosomes paired but not synapsed (Fig 5D), whereas in the control nuclei, the percentage was less than 1% (n = 150). Also, we observed a significant increase in the number of pachytene stage cells and a decrease in the number of diplotene stage cells (Fig 5E). In contrast, the number of germ cells at leptotene and zygotene stages appeared normal (Figs 5E and S4E), suggesting that circSRY functions specifically in the pachytene stage.

Failure of XY body formation usually abolishes meiotic sex chromosome inactivation (MSCI) and increases the expression of X-linked and Y-linked genes in spermatocytes (Fernandez-Capetillo et al, 2003; Royo et al, 2010). To evaluate whether the deficiency of circSRY impacted MSCI, we calculated the average value of gene expression from individual chromosomes based on our RNA-seq analysis in circSRY KO versus control. The average value of all autosomal gene expressions was 1.04 (Fig 5F), which showed that the gene expression level of all autosomal chromosomes from circSRY KO spermatocytes was not different from control. However, the average values of X-linked and Y-linked genes were 2.29 and 2.9, respectively (Fig 5F). These data indicated that circSRY deficiency resulted in the failure of inactivating some of the sex chromosome–linked genes in circSRY KO spermatocytes. In particular, the fold changes of X-linked genes *Rbbp7*, *Cul4b* (Fernandez-Capetillo et al, 2003) and Y-linked gene *Eif2s3y* that were known to be the target of MSCI (Fernandez-Capetillo et al, 2003) were 1.15, 1.47, and 1.33, respectively (Fig 5G). Expression of these genes in 9-wk-old circSRY KO and control spermatocytes was further validated by qRT–PCR, and all three were up-regulated in circSRY spermatocytes (Fig 5H). Taken together, we revealed that the loss of circSRY leads to aberrant X–Y synapsis and impairment of MSCI in pachytene spermatocytes.

### CircSRY acts as a sponge for miR-138-5p

Because cytoplasmic circRNAs could act as miRNA sponges to regulate gene expression indirectly, we predicted potential miRNAs that interacted with circSRY using the web tool miRDB (Liu & Wang, 2019; Chen & Wang, 2020). 33 miRNAs were predicted to potentially interact with circSRY, six of which had more than seven binding sites on circSRY, including well-characterized miR-138-5p (Fig 6A and Table S1) (Hansen et al, 2013). To validate the binding ability of these six miRNAs on circSRY, we constructed luciferase reporter plasmid by inserting circSRY sequence into the 3′UTR of the luciferase coding sequence and performed the experiments in 293T cells. Compared with scrambled miRNA, miR-138-5p, miR-323-5p, and miR-683 reduced the luciferase signals to a greater extent (less than 20%) (Fig 6B). Regarding these three miRNAs, based on published miRNA sequencing data, miR-138-5p was the most abundantly expressed miRNAs in mouse testes, which was six times higher than miR-323-5p, and miR-683 was not detected

(Chiang et al, 2010; Chen et al, 2017). We further conducted absolute quantification of circSRY and miR-138-5p RNA molecules in the spermatocytes and found that the copy numbers of these two RNA molecules were similar, suggesting that miR-138-5p co-existed with circSRY in spermatocytes (Figs 6C and S5A and B). Therefore, we hypothesized that circSRY regulated spermatogenesis mainly by sequestering miR-138-5p.

To test whether circSRY binds to the AGO2-miR-138-5p complex, we performed AGO2 immunoprecipitation on isolated spermatocytes. CircSRY was specifically enriched in AGO2-immunoprecipitated samples (Fig 6D), indicating that circSRY interacts with miRNAs. Next, we established a circSRY over-expressing 293T cell line by transducing a lentiviral vector harboring a circSRY expressing cassette (see the Materials and Methods section) (Fig S5C) and confirmed the correct splicing of circSRY by Sanger sequencing (Fig S5D). We performed RNA pull-down experiment to examine the interaction between miR-138-5p and circSRY in this cell line and found that circSRY-biotinylated probe significantly enriched the miR-138-5p compared with the scramble counterpart (Fig 6E), and the biotin-coupled miR-138-5p captured more circSRY than the control (Fig 6F). Furthermore, in spermatocytes, circSRY probe captured more miR-138-5p than the scramble probe as well (Fig 6G). Taken together, these results suggested that circSRY acted as a sponge for miR-138-5p in spermatocytes.

It has been reported that miR-138-5p directly down-regulates H2AX expression through binding with the 3′UTR region of H2AX mRNA (Fig S5E), inducing chromosomal instability during DNA damage repair (Wang et al, 2011). To test our hypothesis that circSRY regulates H2AX expression via sponging miR-138-5p during spermatogenesis, we conducted luciferase reporter experiments in vitro. Transfection of luciferase reporter containing H2AX sequence together with miR-138-5p mimics showed significantly reduced luciferase signal, whereas co-transfection of circSRY rescued the decrease of luciferase signal (Fig 6H).

We next tried to rescue the reduction of H2AX level and related phenotypes in circSRY KO mice in vivo via testicular efferent duct injection of miR-138-5p antagomir (inhibitor). We chose 6-wk-old circSRY KO mice and injected 5 nmol miR-138-5p antagomir into the left side testis of each mouse and injected PBS into the right side as control. After 2 wk, we euthanized the mice to measure the testis to body weight ratio and sperm count of each mouse. The average ratio of the testis to body weight and the sperm counts in antagomir group showed no increase compared with PBS group (Fig S5F and G). However, the average γH2AX level in antagomir-injected samples was higher than that in PBS-injected samples (Fig S5H). To reveal the source of increased expression of γH2AX, we performed co-immunostaining of γH2AX with DDX4 or SOX9 within antagomir-injected testes of circSRY KO groups. The ratio of DDX4-positive cells to SOX9-positive cells in antagomir-injected testes was comparable to that of the control group (Fig S6A and C), whereas the γH2AX

indicates 20 μm; g, h, scale bar indicates 10 μm). **(E)** Immunofluorescence staining of SYCP3 (red) and SOX9 (green) in seminiferous tubules of *Sry^flox^*; *Amh-Cre* or *Sry^flox^*; *Stra8-Cre* and 9-wk-old control mice. Scale bars indicate 50 μm. **(F)** The ratio of SYCP3-positive cells to SOX9-positive cells within seminiferous tubules from conditional KO mice compared with control mice (**P < 0.01; unpaired *t* test; for *Sry^flox^*; *Amh-Cre*, the total number of tubules reached 25 from three biologically independent experiments; for *Sry^flox^*; *Stra8-Cre*, the total number of tubules reached 50 from three biologically independent experiments). **(G)** The number of SOX9-positive cells within seminiferous tubules from conditional KO mice compared with control mice (ns, not significant; unpaired *t* test; for or *Sry^flox^*; *Amh-Cre*, the total number of tubules reached 25 from three biologically independent experiments; for *Sry^flox^*; *Stra8-Cre*, the total number of tubules reached 50 from three biologically independent experiments). Data information: data are presented as mean ± SEM. Source data are available for this figure.

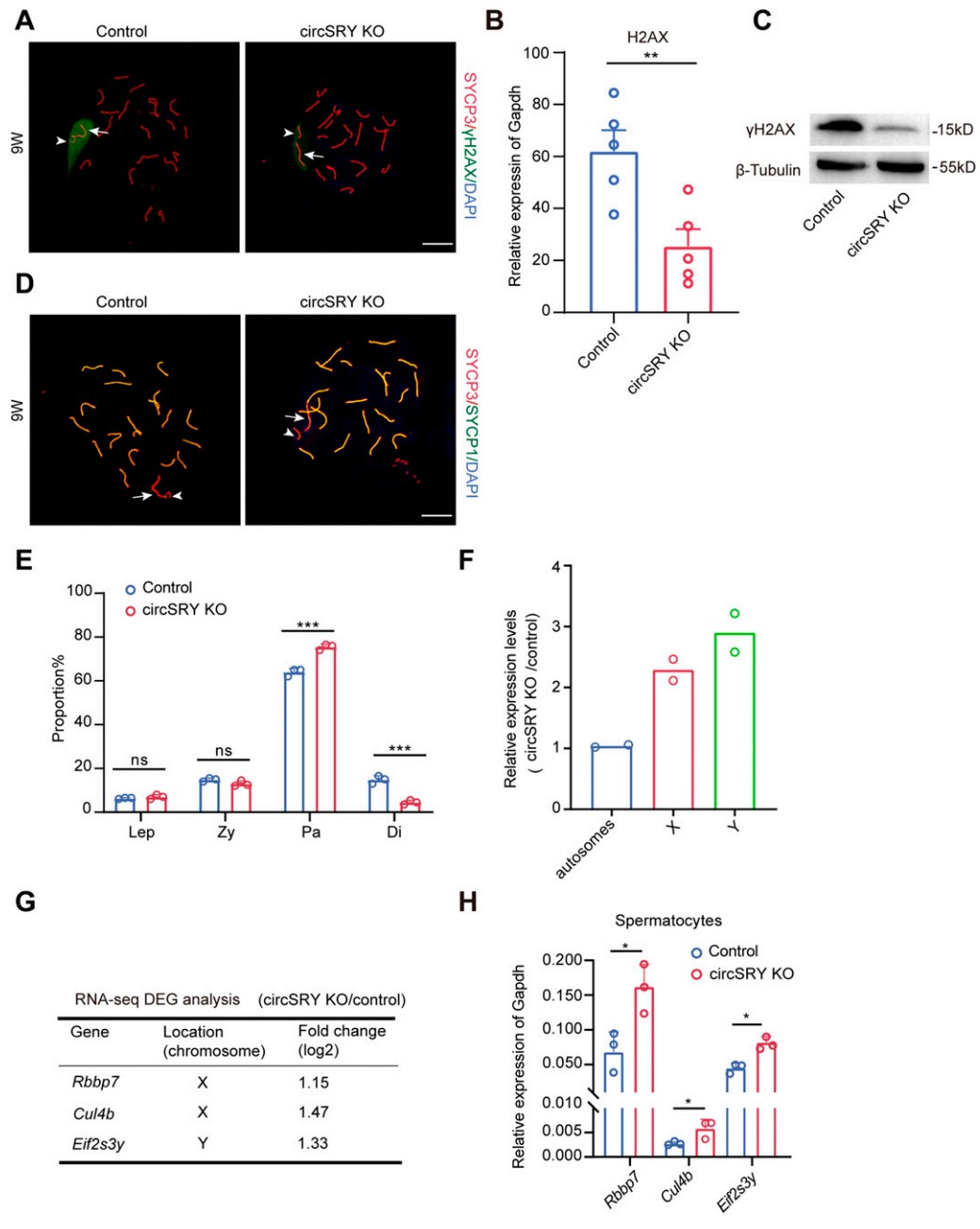

**Figure 5. Loss of circSRY led to aberrant X–Y synapsis.**
**(A)** Immunofluorescence staining of SYCP3 (red) and γH2AX (green) in 9-wk-old control and circSRY KO spermatocytes at early pachytene stage. Arrowheads indicate the Y chromosomes; arrows indicate the X chromosomes. Scale bar indicates 10 μm. **(B)** Expression of H2AX mRNA in 9-wk-old control and circSRY KO mice spermatocytes (**P < 0.01; unpaired t test; n = 5 biologically independent experiments). **(C)** Representative image of Western blot results of γH2AX in control and circSRY KO mice spermatocytes (n = 3 biologically independent experiments). **(D)** Immunofluorescence staining of SYCP3 (red) and SYCP1 (green) in 9-wk-old control and circSRY KO spermatocytes at early pachytene stage. Arrowheads indicate the Y chromosomes; arrows indicate the X chromosomes. Scale bar indicates 10 μm. **(E)** The proportion of leptotene, zygotene, pachytene, and diplotene spermatocytes from 9-wk-old circSRY KO mice compared with control mice testes (***P < 0.001; ns, not significant; unpaired, t test). Total 400 cells (300 cells of pachytene stage, 100 cells of the other stages) were counted from three biologically independent experiments. **(F)** Relative gene expression levels on sex chromosomes and autosomes as determined by RNA-seq (n = 2 biologically independent experiments). **(G)** Fold change of X-linked genes, Rbbp7, Cul4b; Y-linked genes, Eif2s3y, as determined by RNA-seq analysis on circSRY KO versus control spermatocytes (n = 2 biologically independent experiments). **(H)** Validation of expression of X-linked genes, Rbbp7, Cul4b; Y-linked genes, Eif2s3y from control versus 9-wk-old circSRY KO spermatocytes (*P < 0.05; ns, not significant; unpaired, t test; n = 3 biologically independent experiments). Gapdh was used as reference gene. Data information: data are presented as mean ± SEM.
Source data are available for this figure.

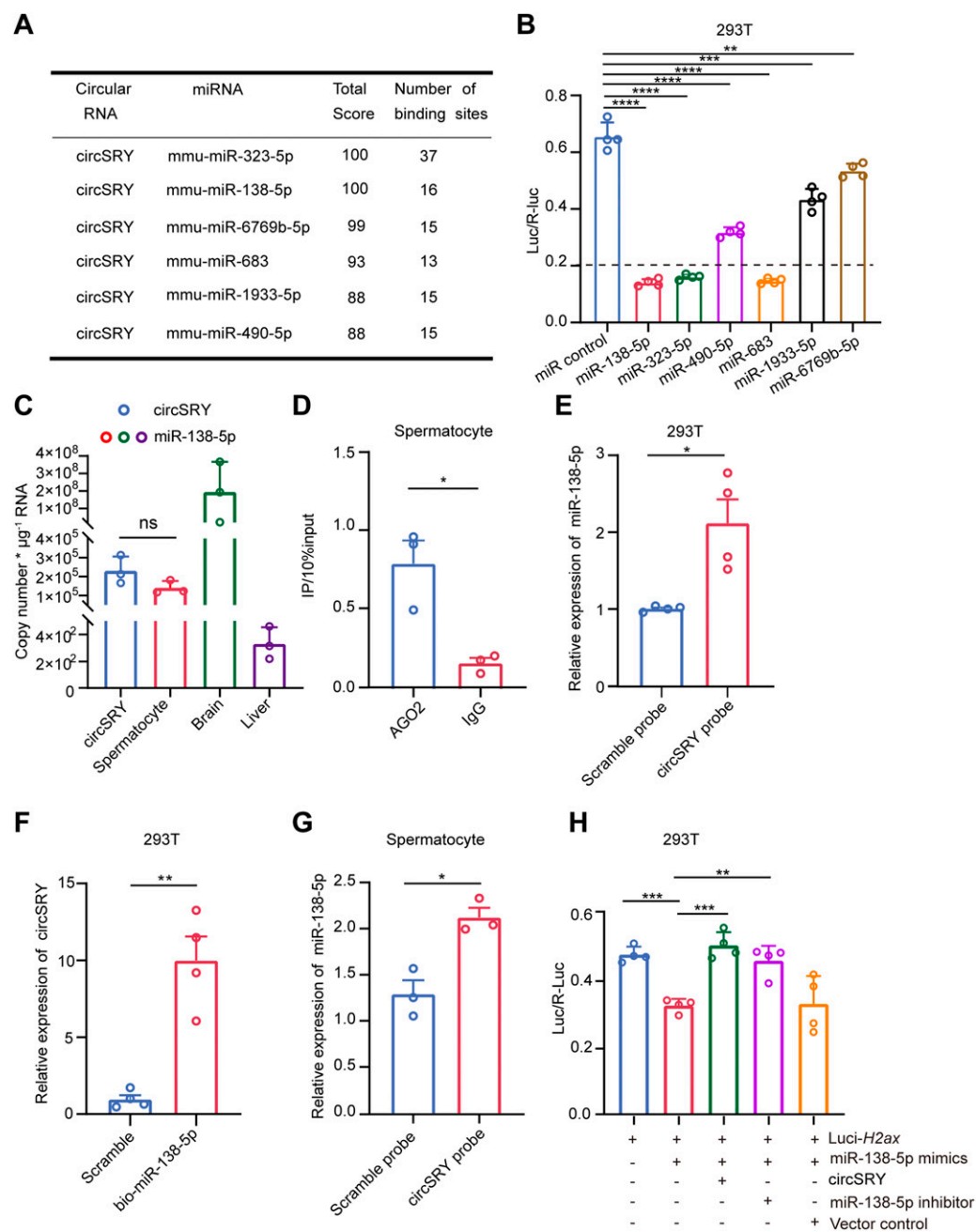

**Figure 6. CircSRY enhanced the expression of H2AX by sponging miR-138-5p in vitro.**
**(A)** Table of six miRNAs that have more than seven potential binding sites for circSRY. **(B)** Luciferase reporter assay showing diverse binding capacities of six miRNAs on Luc-circSRY (****$P$ < 0.0001, ***$P$ < 0.001, **$P$ < 0.001, unpaired $t$ test; n = 4 biologically independent experiments). **(C)** Absolute quantification of circSRY and miR-138-5p RNA molecules in spermatocyte. Mouse brain or liver tissue RNAs were used as control, separately (ns, not significant; unpaired $t$ test; n = 3 biologically independent experiments). **(D)** RIP results showing the circSRY enrichment using AGO2 antibody versus IgG antibody in vivo (*$P$ < 0.05; unpaired $t$ test; n = 3 biologically independent experiments). **(E)** RNA pull-down enrichment of miR-138-5p using circSRY-biotinylated probe or scramble probe in 293T cell line. (*$P$ < 0.05; unpaired $t$ test; n = 4 biologically independent experiments). **(F)** RNA pull-down enrichment of circSRY using biotin-coupled miR-138-5p in 293T cell line. (**$P$ < 0.01; unpaired $t$ test; n = 4 biologically independent experiments). **(G)** Expression of miR-138-5p pulled down by circSRY-biotinylated probe or scramble probe in spermatocytes (*$P$ < 0.05, unpaired $t$ test; n = 3 biologically independent experiments). **(H)** Luciferase reporter assay of circSRY on sponging miR-138-5p. MiR-138-5p inhibitor and empty vector were used as positive and negative control, separately (**$P$ < 0.01, ***$P$ < 0.001; unpaired $t$ test; n = 4 biologically independent experiments). *Gapdh* was used as reference gene. Data information: data are presented as mean ± SEM.
Source data are available for this figure.

levels were increased in antagomir-injected testes (Fig S6B and D), implying the source of increased γH2AX came from the elevated expression of γH2AX within germ cells. This suggested that antagomir injection led to an increase in γH2AX level intensity in the testis. However, the increased expression of γH2AX did not lead to an increase in testis weight and sperm count, which may be because of the invasiveness of the in vivo experimental procedures or the complexity of meiotic process.

In summary, our results suggest that circSRY acts as a sponge to competitively bind miR-138-5p, and the loss of circSRY leads to reduced γH2AX level and defected meiosis, which further impairs spermatogenesis in adult mice.

# Discussion

Recent studies have identified thousands of circRNAs, and many of them have important biological functions in various tissues and organs. For example, circRNA participates in the regulation of innate immune response by forming an intramolecular double-stranded structure when cells are infected by viruses (Liu et al, 2019). CircPan3 participates in the activation of the IL-13/IL-13Rα1 pathway and the downstream Wnt/b-catenin pathway, which maintained the self-renewal of intestinal stem cells (Zhu et al, 2019). Here we show that a circular transcript exhibits regulatory function in the male reproductive system. More interestingly, this circRNA originates from the sex-determining gene *Sry*, which initiates the male germ cell development in the first place. In this study, we showed that circSRY was mainly detected in meiotic cells and Rs. The loss of circSRY led to defective spermatogenesis, specifically causing primary spermatocyte apoptosis. Deletion of circSRY decreased the expression of H2AX and displayed aberrant X–Y synapsis at the pachytene stage, followed by MSCI defect. We performed RNA pull-down experiment and luciferase reporter assay, demonstrating that circSRY binds to miR-138-5p in spermatocytes and is capable of enhancing H2AX expression by sequestering miR-138-5p in 293T cell lines.

We tried to rescue the phenotypes in circSRY KO mice in vivo by injecting the left side testis of each mouse with miR-138-5p antagomir and the right side with PBS as control. We did not observe a phenotypic recovery in antagomir-injected testis; this may be due to the complexity of the reproductive system and the invasive nature of the injection experiment, which may have caused damage to the testis to some extent. A better model of spermatogenesis is needed to verify the relationship between circSRY, miR-138-5p, and γH2AX. Also, compared with *H2ax* KO leading to infertility (Celeste et al, 2002), circSRY KO reduced H2AX level, leading to a lower number of sperm, which is a much milder phenotype that is more challenging to demonstrate clear rescue in vivo. Our study indicates that the loss of circSRY leads to low H2AX level in spermatocytes. H2AX phosphorylation plays key roles in the initiation of heterochromatinization in the sexual body (Fernandez-Capetillo et al, 2003). Moreover, the phosphorylation of H2AX on the unsynapsed sexual body is independent of SPO11-mediated DSB (Mahadevaiah et al, 2001) and ATM kinase but dependent on ATR kinase (Turner et al, 2004; Bellani et al, 2005). Recent research

shows that MDC1 deletion leads to the failure of spreading of γH2AX to the sexual body; the γH2AX signal is confined to the unsynapsed regions of sex chromosomes (Ichijima et al, 2011). Another study shows that MDC1 is dispensable for the synapsis of autosomes and plays only a minor role in X–Y synapsis (Testa et al, 2018), suggesting that γH2AX dominates over MDC1 in the process of XY body formation. In our study, RNA-seq analysis showed a decreased expression of MDC1 (lgFC = -2.39) which is the key regulator for the initiation of MSCI (Ichijima et al, 2011). Besides MDC1, two other important regulators of MSCI, ATR and TOPBP1 (Perera et al, 2004; Reini et al, 2004) also had a reduced expression compared with control (lgFC = -1.92; lgFC = -2.66). The underlying mechanism of such changes demands further exploration.

Mouse *Sry* gene is capable of producing linear RNAs at the embryonic stage and circular transcripts in the adult testis. It was proposed that circSRY is generated from a linear RNA precursor containing long palindromic repeats, which is transcribed from a distal promoter (Dolci et al, 1997). In our study, the deletion of splicing acceptor did not interfere with the sex determination. However, a hidden exon was found in the long palindromic repeats region that maintained the stability of SRY protein during sex determination (Miyawaki et al, 2020). Moreover, the second exon of *Sry* uses the same splicing site as circSRY to generate coding mRNA, whereas circSRY uses it as the splicing donor to back-splice. Further investigation would be needed to reveal how the choice of these splicing events was regulated. In addition to *Mus musculus*, transcripts of Sry were also detected in the testes of *M. musculus* domesticus and *Mus spretus*. Considering that the splicing donor site was also conserved in *Rattus norvegicus* (Miyawaki et al, 2020), we speculate that alternative splicing occurs in rats as well.

Human *SRY* is also expressed in adult testes. Different from the circular transcript in adult mouse testis, the human *SRY* transcript is a linear and polyadenylated transcript (Sinclair et al, 1990), and the presence of circular SRY has not been reported. The function of human *SRY* transcript in adult testes is very curious, and the elucidation of which may help us better understand male development and spermatogenesis in humans. Although human *SRY* transcript in testes might regulate germ cell development using a different mechanism, the sex-determining gene also plays a role in adult testes could be a common theme in multiple mammalian species.

In addition to the testis, circSRY was also detected in the mouse brain during embryonic development, and its expression was diminished after birth (Sinclair et al, 1990). Interestingly, miR-138-5p is also expressed during the embryonic stage of mouse brain development (Obernosterer et al, 2006). It would be really interesting to explore the potential interactions between circSRY and miR-138-5p and to understand their functions in brain development. Considering that *Sry* only exists in males, if circSRY or SRY protein (Mayer et al, 2000) plays a role in the mouse brain, it could potentially contribute to the sex dimorphisms of neural systems between males and females. The mouse models we generated in this study will be very helpful for these future studies.

Taken together, our study complements *Sry*'s role in germ cell development, revealing its significance in male germ cell development. CircSRY helps safeguard the inactivation of sex chromosomes during the pachytene stage, the key process to complete

meiosis and produce sperm. Upon fertilization, the sperm carrying the Y chromosome contributes Y to the embryo which will develop into a male and grow testicles. The cycle of *Sry* expression and dual forms and functions suggests a mechanism to ensure the preservation of vulnerable Y chromosomes in evolution.

# Materials and Methods

## Animals

The mice used in this research were of C57BL/6 background. All animals were maintained at 24°C and 50–60% humidity under a 12: 12 h light/dark cycle and with ad libitum access to food and water. *Sry*$^{flox}$ mice were generated in Jackson Laboratory. Heterozygous *Stra8-Cre* and *Amh-Cre* mice with the background of 129/SvEv were provided by Prof. Gao of the Institute of Zoology, University of Chinese Academy of Sciences. All mice studies were carried out in accordance with the principles approved by the Institutional Animal Care and Use Committee at the Institute of Zoology, Chinese Academy of Sciences.

## Establishment of mutant mice by CRISPR-Cas9–based genome editing

CircSRY knockout mice were established by microinjection of Cas9 mRNA and sgRNAs into zygotes (Qin et al, 2016). All RNAs prepared for microinjection were generated through in vitro transcription. Briefly, Cas9 mRNAs and sgRNAs were mixed properly in a 1:2 ratio and injected into zygotes. The injected embryos were transferred to pseudopregnant females. To genotype mutant mouse lines, the genomic sequences were amplified by PCR, followed by Sanger sequencing. All oligonucleotides used were listed in Table S2.

## Establishment of *Sry*$^{flox}$ mice by CRISPR-Cas9 system

We used single-stranded oligodeoxynucleotides (ssODN) to establish knock-in mouse mutants (Qin et al, 2016). Establishment of *Sry*$^{flox}$ KI mice went through two rounds of microinjections. The first microinjection contained Cas9 mRNA, and ssODN (V5-loxP) targeted the 3′ end of *Sry* exon1 (V5-TGA-loxP). The injected embryos were transferred to pseudopregnant females. The correct male mouse line was determined by Sanger sequencing of genomic sequence. Sperm from *Sry*-V5-loxP male mouse line was collected for single sperm microinjection of WT oocytes. The second microinjection contained Cas9 mRNA, and ssODN (5′-loxP) targeted the 5′ upstream site of the *Sry* exon1 start codon within the *Sry*-V5-loxP zygotes. The correct mouse line (loxP-*Sry*-V5-loxP) was again confirmed by Sanger sequencing of genomic sequence. All oligonucleotides and primer sequences used in this study are listed in Table S2.

## Conditional knockout of *Sry* in germ cells or in Sertoli cells

Conditional mutant mice were obtained by crossing the *Sry*$^{flox}$ male mice with *Stra8-Cre* heterozygous mice or *Amh-Cre* heterozygous

adult female mice to generate newborns. The tail of the newborn was used for genotyping.

## Genotyping of mice and sexing

All mice were genotyped with the tail DNA. Chromosomal sex was detected by amplifying the Y-linked gene *Uty*. Phenotypic sex was determined by examination of the external genitalia and the presence or absence of mammary glands. Primers used in this study are listed in Table S2.

## RNA preparation and real-time PCR

Nuclear and cytoplasmic RNAs were extracted using Norgen's Cytoplasmic & Nuclear RNA Purification Kit (Cat: 21000; Norgen Biotek Corp.). RNAs were extracted using Trizol reagent (T9424; Life Technologies). cDNAs were synthesized using the Hifair III first strand cDNA synthesis reagent (11141ES60; Yeasen Company) with 500 ng of total RNA. cDNAs were amplified with Hieff qPCR SYBR Green Master Mix (11202ES08; Yeasen Company) and quantified with Roche LightCycler 480 system. For RNase R treatment, the experiment was performed by incubation of 3 µg of RNA with 6 U µg$^{-1}$ of RNase R (RNA07250; Epicenter) for 25 min at 37°C. The expression level of each gene was presented relative to *Gapdh* or U6 expression. The primer sequences used are shown in Table S2.

## RNA pull-down

Biotinylated circSRY probes and 5′bio-miRNA mimics were synthesized by RiboBio. Testes were ground and incubated in lysis buffer (50 mM Tris–HCl, pH 7.4, 150 mM NaCl, 2 mM MgCl$_2$, 1% NP40, 20 U/ml RNase inhibitor [R0102-2KU; Beyotime Biotechnology]) on ice for 1 h. The lysates were then incubated with the biotinylated probes at RT for 4 h, followed by adding streptavidin C1 magnetic beads (65001; Invitrogen) as part of binding reaction and continued to incubate at 4°C overnight. On the second day, the beads were washed briefly with wash buffer (0.1% SDS, 1% Triton X-100, 2 mM EDTA, 20 mM Tris–HCl, and 500 mM NaCl) five times. The bound RNA in the pull-down was further extracted for purification and qRT–PCR. The sequences of circSRY probe are shown in Table S3.

## RNA-binding protein immunoprecipitation

RIP experiments were performed with primary spermatocytes which separated from adult mice testes and homogenized into a single-cell suspension in ice-cold PBS. After centrifugation, the pellet was resuspended in RIP lysis buffer. Magnetic beads were incubated with a 5 µg antibody against AGO2 (10686-1-AP; Proteintech Inc.) or IgG (10283-1-AP; Proteintech Inc.) at RT. The tissue lysates were then incubated with the bead–antibody complexes overnight at 4°C. RNAs were extracted by Trizol reagent and reverse-transcribed after proteinase K treatment.

## RNA-seq and data analysis

RNA sequencing reads were aligned to mouse reference sequence GRCm39 using STAR (2.7.0f). Read counts and FPKM (transcripts per

kilobase million) were counted using RSEM (1.3.2). Differential expression genes (FDR < 0.05, $\log_2$ [fold change] [$\log_2$ FC] ≥ 1 or ≤ −1) were calculated by edgeR. The differential expression genes were carried out by GO pathway analysis and Kyoto Encyclopedia of Genes and Genomes analysis by clusterProfiler (4.2.2) and org.Hs.eg.db (3.10.0) packages. The volcano plot was calculated by ggplot2 (3.3.5). Spearman's rank correlation analysis and heat map were calculated by heatmap (1.0.12). Colors represent the Z-score derived from the $\log_2$-transformed FPKM data.

### Construction of 293T-circSRY cell line

The empty vector was purchased from Geneseed Bio (GS0108), termed pLC5. This plasmid was linearized with XhoI-BamHI. The complete circSRY sequence was amplified with XhoI-BamHI cloning site and inserted into pLC5. The final plasmid was named after pLC5-circSRY. pLC5-circSRY was packaged with lentivirus and transduced into 293T cell line. The expression of circSRY was testified, and the splicing acceptor site was confirmed after 3 d of puromycin selection.

### Dual-luciferase reporter assay

The full-length sequence of circSRY or the 3′UTR of H2AX was inserted into the 3′ UTR of pMIR-Report Luciferase vector (gifted by Yu Wang lab, State Key Laboratory of Stem Cell and Reproductive Biology, Institute of Zoology, Chinese Academy of Sciences). Co-transfection of pMIR-Report Luciferase vector, miRNA mimic (purchased from RiboBio), and pLC5-circSRY was conducted using Lipofectamine 2000 (11668027; Invitrogen). After 48 h, the luciferase activities were measured using a dual-luciferase reporter assay kit (E1910; Promega). Firefly luciferase activity was normalized to the Renilla luciferase signal.

### Western blotting

Proteins obtained from spermatocytes were separated by gel electrophoresis (SDS–PAGE) and transferred to a PVDF membrane. The PVDF membrane was incubated with primary antibodies against γH2AX (1:200, 05-636; Millipore), β-Tubulin (1:1,000, 30301ES40; Yeasen Company) at 4°C overnight, and HRP-labeled secondary antibody for 1 h at 37°C subsequently. Images were captured using ECL Western Blotting Substrate (32106; Thermo Fisher Scientific).

### TUNEL assay

The TUNEL assay was performed by TUNEL BrightRed Apoptosis Detection Kit (A113-01; Vazyme). Briefly, sections were permeabilized by protein K and labeled with rTdT reaction mix for 1 h at 37°C. The reaction was stopped by 1× PBS. After washing with PBS, the sections were incubated in 2 $\mu$g/ml of DAPI (D1306; Molecular Probes) for 10 min. Sections were mounted on slices with Fluorescence Mounting Medium (F4680; Sigma-Aldrich). Images were obtained using a laser scanning confocal microscope LSM780 (Carl Zeiss).

### Flow cytometry

The cell suspension obtained from testes was digested by collagenase IV and trypsin for 5 min into a large cell suspension. To analyze DNA ploidy, the cells were incubated with Hoechst for 15 min and filtered before being subjected to flow cytometry. The results were analyzed using a FACSCalibur system (BD Biosciences). The sorting gate strategy of flow cytometry was optimized according to the reference (Gaysinskaya et al, 2014).

### Sperm count and fertility

The epididymis tail of the mouse was taken out, cut into pieces with ophthalmic scissors, placed in a 37°C water bath for 15 min, and counted via microscope. 2–3-mo-old circSRY and control C57BL/6J male mice were housed with control C57BL/6J males (2–3-mo-old), which were proved to have normal fecundity. Copulatory plugs were monitored daily, and plugged females with visibly growing abdomen were moved to separate cages for monitoring pregnancy. The mating process lasted for 4 mo. The number of pups (both alive and dead) was counted on the first day of life.

### Tissue collection and histological analysis

Testes from control and circSRY KO mouse were dissected immediately after euthanasia, then immediately fixed with 4% PFA for 24 h; after storing in 70% ethanol and embedding in paraffin, 5 $\mu$m-thick sections were prepared using a rotary microtome (Leica) and mounted on glass slides. After deparaffinization, sections were stained with hematoxylin–eosin for histological analysis.

### Immunohistochemical analysis

IHC analysis of tissue from at least three males was performed using a Vectastain ABC (avidin–biotin–peroxidase) kit (Vector Laboratories). Antibodies against DDX4 (1:500) were purchased from Abcam (ab13480). SOX9 antibodies (1:100) were purchased from Chemicon (AB5535). PLZF antibodies (1:200) were purchase from Invitrogen (MA5-15667). After staining, the sections were examined with a Nikon microscope, and images were captured with Nikon DS-Ri1 CCD camera.

### Periodic acid–Schiff/hematoxylin stain

Testes sections from control and circSRY KO mouse were placed in periodic acid solution (1%) for 10 min, then immediately washed at least three times with distilled water. Next, sections were covered with Schiff's reagent for 20 min, followed by washing three times with distilled water. Then sections were covered with hematoxylin for 1 min and differentiated and blued. Finally, sections were mounted in DPX and prepared for observation.

### Immunofluorescence analysis

After deparaffinization and antigen retrieval, 5% bovine serum was used to block sections at RT for 1 h, and specific primary antibody SYCP1 (1:200, ab15087; Abcam), γH2AX (1:400, 05-636; Millipore), PLZF

(1:100, AF2944; R&D), SOX9 (1:500, AB5535; Millipore), STRA8 (1:200, ab49405; Abcam), SYCP3 (1:200, ab15093; Abcam), PIWILI1 (1:200, A2150; Abclonal), F-ACTIN (1:200, Ab130935), DDX4 (1:200, Ab13840) was used to incubate with sections at RT or overnight at 4°C. After washing the sections three times, the slides were incubated with the corresponding secondary antibody, fluorescent dye–conjugated FITC (1:150, 115-095-003; Jackson) or TRITC (1:150, 115-025-003; Jackson) for 1 h at RT (avoiding the light). DAPI was used to stain the nucleus. All images were captured with confocal microscopy (Leica TCS SP8).

### Preparation of synaptonemal complex

Seminiferous tubules were collected from dissecting testes and washed with 1× PBS. Hypo extraction buffer was used to incubate within seminiferous tubules for 30 min, followed by disrupting within 0.1 M sucrose liquid to form a single-cell suspension. The cell suspension was mounted on slides treated with 1% PFA. Slides were air-dried in a humidified box for at least 6 h. After washing with 0.04% Photo-Flo (Equl, 1464510), the slides were stained for SYCP3, SYCP1 (1:200; Abcam), $\gamma$H2AX (1:200; Millipore). Incubation with the specific antibody was conducted at RT for 30 min after antibody dilution buffer treatment. After washing three times in 1× Tris buffer, saline (TBS), blocking was conducted with 1× antibody dilution buffer at 4°C overnight. After washing three times with cold 1× TBS buffer, the corresponding secondary antibody and the fluorescent dye–conjugated TRITC were incubated with sections for 3 h at 37°C. All images were captured with a confocal laser scanning microscope (Leica TCS SP8).

### Testicular efferent duct injection

The mice were anesthetized, and the testes were pulled out from the abdominal cavity. 10 $\mu$l of solution (5 nmol miR-138-5p antagomir, miR3CM001; Ruibo Company) was injected into the rete testis using with a glass capillary under a stereomicroscope. Mice were given 2 wk to recover.

### Data and statistical analysis

All images were processed with Photoshop CS6 (Adobe). All statistics were analyzed using Prism software (GraphPad Software). All experiments were performed in triplicate (except for RNA-seq, transfection of the pLC5-cirSRY assay, and percentage of cell purity analysis), and results were confirmed in at least three independent experiments. Three to five control or mutant testes were used for immunostaining. The quantitative results were presented as the mean ± SEM. The significant difference was evaluated with the $t$ test. $P$-value < 0.05 was considered significant.

### Cell lines and primary cell types

Human embryonic kidney 293T cells were purchased from the Peking Union Medical College Cell Culture Center and cultured in DMEM (Gibco, Life Technologies) supplemented with 10% FBS, penicillin (100 U/ml⁻), and streptomycin (100 mg/ml⁻) at 37°C with 5% $CO_2$. Sertoli cells were isolated from P20 mouse testes, which

were digested for 10 min at 37°C with collagenase type IV (2 mg/ml; C5138; Sigma-Aldrich) and further digested with trypsin (10 mg/ml) and DNase I (40 mg/ml) in the DMEM for 15 min at 37°C. The digest was passed through a 75-micropore nylon mesh and washed with DMEM and cultured in six-well plates. The cells were kept in DMEM supplemented with 10% FBS, penicillin (100 U/ml⁻), and streptomycin (100 mg/ml⁻) at 37°C with 5% $CO_2$.

## Data Availability

To review GEO accession GSE197971, go to https://www.ncbi.nlm.nih.gov/geo/query/acc.cgi?acc=GSE197971.

## Supplementary Information

## Acknowledgements

We are grateful to Chenrui An for their comments and editing of the article and all members of Fei Gao laboratory for technical support. This work was supported by the National Key Research and Development Program of China (2018YFE0201102, 2019YFA0110000, 2018YFA0107703, and 2016YFA0101402), Strategic Priority Research Program of the Chinese Academy of Sciences (XDA16010503), Strategic Collaborative Research Program of the Ferring Institute of Reproductive Medicine, Ferring Pharmaceuticals, Chinese Academy of Sciences (FIRMD181101), National Natural Science Foundation of China (31722036, 81773269 and 32001062).

### Author Contributions

Y Song: resources, data curation, software, formal analysis, investigation, methodology, project administration, and writing—original draft, review, and editing.
M Chen: investigation, project administration, and writing—original draft, review, and editing.
Y Zhang: data curation and investigation.
J Li: data curation.
B Liu: data curation.
N Li: data curation, software, and investigation.
M Chen: data curation, software, and investigation.
M Qiao: software.
N Wang: software.
Y Cao: data curation.
S Lu: software.
J Chen: conceptualization and data curation.
W Sun: software, supervision, funding acquisition, validation, and writing—original draft, review, and editing.
F Gao: conceptualization, supervision, project administration, and writing—original draft, review, and editing.
H Wang: conceptualization, supervision, funding acquisition, validation, and writing—original draft, review, and editing.

## Conflict of Interest Statement

The authors declare that they have no conflict of interest.

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
