## [Reviewer comments · Life Science Alliance]

Life Science Alliance

Loss of circSRY reduces γ H2AX level in germ cells and impairs mouse spermatogenesis

Yanze Song, Min Chen, Yingfan Zhang, Jiayi Li, Bowen Liu, Na Li, Min Chen, Miaomiao Qiao, Nan Wang, Yuanwei Cao, Shan Lu, Jian Chen, Wen Sun, Fei Gao, and Haoyi Wang

DOI: <https://doi.org/10.26508/lsa.202201617>

Corresponding author(s): Haoyi Wang, Chinese Academy of Sciences and Fei Gao, Institute of Zoology, Chinese Academy of Sciences

Review Timeline:

Submission Date:	2022-07-20
Editorial Decision:	2022-07-20
Revision Received:	2022-10-25
Editorial Decision:	2022-10-26
Revision Received:	2022-10-30
Accepted:	2022-11-07

Transaction Report:

Please note that the manuscript was previously reviewed at another journal and the reports were taken into account in the decision-making process at Life Science Alliance.

Referee #1 Review

Report for Author:

I raised two major concerns in the initial review: 1) co-localization of circSRY with miR-138-5p in spermatocytes; 2) a lack of rescue experiments to validate the proposed sponge function. The first concern was addressed adequately, but the second remains because the experiments performed were highly questionable. The antagomir strategy is not reliable at all. They injected 10 microliters of solution into the testis, which for sure would cause damage to the spermatogenic cells due to increased intratubular pressure.

Referee #3 Review

Report for Author:

In this manuscript, Song and colleagues examine the role of the circular form of the Sry RNA (circSry), which is a master regulator of mammalian sex (testis) determination. They find that, while it is not required for somatic sex determination, circSry is required in meiotic germ cells, as circSry KO mice exhibit reduced sperm count, likely due to increased apoptosis and loss of spermatocytes. They propose a mechanism in which circSry acts as a sponge for miR-138-5p to regulate H2ax expression. Overall, the studies are rigorously performed and the data is clearly presented in detail. The strength of the study is in its novel examination of the circular version of Sry and finding a germ-cell-specific role for this version of the RNA. These findings are of interest to both the general molecular biology, for its study of circular RNA and gene regulation, and to the reproductive biology field, for its examination of a new role for Sry. However, there are some shortcomings in the manuscript, the main one being that the link between circSry, miR-138-5p, and γ H2Ax is not strongly demonstrated, which weakens support for the model proposed by the authors.

1. The antagomir experiments in Fig. EV5 are rather limited, especially when compared to similar analyses in Figs. 4 and 5. In particular, it is unclear what the source of the increased γ H2AX levels is. Is it due to more spermatocytes? More expression within the same number of spermatocytes? More DNA damage in general (since γ H2AX also can be due to general DNA damage and is not specific to meiosis)? Furthermore, it is unclear why only a subset of testes were chosen in Fig. EV5H (6

testes), when the authors had many samples from which to choose. If there was truly a significant increase in γ H2AX expression, a larger sample size would have strengthened this analysis; conversely, if they were to use all the samples and would find a large variability that results in a non-significant difference, then the current limited analysis appears to be misleading. Finally, are there any positive controls available to verify that the antagomir actually worked to reduce miR-138-5p activity? Overall, there are several major concerns with the antagomir assays. At a minimum, any claims about links between H2ax and miR-138-5p should be softened.

2. In Figure 2F, the authors need to quantify cell death (TUNEL+ cells).

3. In Fig. 2I, 3B, and EV3B (and elsewhere), the y-axis label units are misleading for the sperm counts since it jumps from 0 to 5×10^6 . This scale is somewhat misleading in terms of the magnitude of change.

4. Was it surprising in Fig. 4C that there was no reduction in haploid (1N) cells, since there was a reduction in 4N cells and a reduction in sperm count in the CircSry cKO mice? These findings don't quite add up.

5. Are the gene expression changes in Fig. 5G statistically significant?

6. Statistics should be performed and significant differences should be denoted in Fig. 6B.

7. Detailed information on mouse strains used should be included in the Methods section.

8. Minor point: GAPDH should be labeled as Gapdh in qRT-PCR graphs, since it refers to RNA and not protein.

9. Minor point: There are still many remaining minor typographical and grammatical errors in the manuscript. These likely can be addressed during copyediting.

July 20, 2022

Re: Life Science Alliance manuscript #LSA-2022-01617-T

Prof. Haoyi Wang
Chinese Academy of Sciences
Institute of Zoology
Beijing 100101
China

Dear Dr. Wang,

Thank you for submitting your manuscript entitled "Loss of circSRY impairs mouse spermatogenesis via reducing H2AX level in germ cells" to Life Science Alliance. We invite you to re-submit the manuscript, revised to address the following points:

- Due to the concerns with the antagomir assays, the claims made regarding links between H2AX and miR-138-5p should be toned down.
- Address Reviewer 3's comments.

When submitting the revision, please include a letter addressing the Reviewers' comments point by point.

Thank you for this interesting contribution to Life Science Alliance. We are looking forward to receiving your revised manuscript.

Sincerely,

- A letter addressing the reviewers' comments point by point.
- An editable version of the final text (.DOC or .DOCX) is needed for copyediting (no PDFs).
- High-resolution figure, supplementary figure and video files uploaded as individual files: See our detailed guidelines for preparing your production-ready images, <https://www.life-science-alliance.org/authors>
- Summary blurb (enter in submission system): A short text summarizing in a single sentence the study (max. 200 characters including spaces). This text is used in conjunction with the titles of papers, hence should be informative and complementary to the title and running title. It should describe the context and significance of the findings for a general readership; it should be written in the present tense and refer to the work in the third person. Author names should not be mentioned.
- By submitting a revision, you attest that you are aware of our payment policies found here: <https://www.life-science-alliance.org/copyright-license-fee>

B. MANUSCRIPT ORGANIZATION AND FORMATTING:

Editor:

I am happy to inform you that Dr. Sawey is interested in your findings, and would like to publish this manuscript at LSA pending the following revisions:

Due to the concerns with the antagomir assays, the claims made regarding links between H2AX and miR-138-5p should be toned down.

Response: We thank you for your valuable comment and will address the reviewers' concerns the best way possible.

We did not observe a phenotypic recovery in antagomir assays, our current model only partially explained the relationship between circSRY, miR-138-5p, and γ H2AX, thus we have revised the summary in the last paragraph to “In summary, our results suggest that circSRY acts as a sponge to competitively bind miR-138-5p *in vivo*, and the loss of circSRY leads to reduced γ H2AX level and defected meiosis, which further impairs spermatogenesis in adult mice.” We also replaced the manuscript title with “Loss of circSRY reduces γ H2AX level in germ cells and impairs mouse spermatogenesis” for a more accurate statement.

Referee #3:

In this manuscript, Song and colleagues examine the role of the circular form of the Sry RNA (circSry), which is a master regulator of mammalian sex (testis) determination. They find that, while it is not required for somatic sex determination, circSry is required in meiotic germ cells, as circSry KO mice exhibit reduced sperm count, likely due to increased apoptosis and loss of spermatocytes. They propose a mechanism in which circSry acts as a sponge for miR-138-5p to regulate H2ax expression. Overall, the studies are rigorously performed and the data is clearly presented in detail. The strength of the study is in its novel examination of the circular version of Sry and finding a germ-cell-specific role for this version of the RNA. These findings are of interest to both the general molecular biology, for its study of circular RNA and gene regulation, and to the reproductive biology field, for its examination of a new role for Sry. However, there are some shortcomings in the manuscript, the main one being that the link between circSry, miR-138-5p, and

γ H2Ax is not strongly demonstrated, which weakens support for the model proposed by the authors.

Response: Thanks for your time reviewing our manuscript. Your encouraging comment is greatly appreciated. Due to the unsuccessful phenotypic recovery in antagomir assays, our current model only partially explained the relationship between circSRY, miR-138-5p, and γ H2AX. Thus, we agree with you about the shortcomings in the manuscript which is also mentioned by the editor and toned down the claims regarding links between H2AX and miR-138-5p. We have revised the summary in the last paragraph to “In summary, our results suggest that circSRY acts as a sponge to competitively bind miR-138-5p *in vivo*, and the loss of circSRY leads to reduced γ H2AX level and defected meiosis, which further impairs spermatogenesis in adult mice.” We also changed the manuscript title into “Loss of circSRY reduces γ H2AX level in germ cells and impairs mouse spermatogenesis” for a more accurate statement.

1. The antagomir experiments in Fig. EV5 are rather limited, especially when compared to similar analyses in Figs. 4 and 5. In particular, it is unclear what the source of the increased γ H2AX levels is. Is it due to more spermatocytes? More expression within the same number of spermatocytes? More DNA damage in general (since γ H2AX also can be due to general DNA damage and is not specific to meiosis)?

Response: We thank the reviewer for this comment. We performed co-immunostaining of DDX4 with γ H2AX or SOX9 in miR-138-5p antagomir-injected testes of circSRY KO mice to verify the source of increased γ H2AX. The ratio of DDX4 positive cells to SOX9 positive cells in antagomir-injected testes was comparable to that of the control group. However, the γ H2AX level was increased in antagomir-injected testes, implying the increased γ H2AX came from the elevated expression of γ H2AX within germ cells. The data were shown below.

Elevated expression of γ H2AX was detected in antagomir-injected testis

(A) Representative image of immunofluorescence staining of SOX9 (red), MVH (green) and DAPI (blue) in antagomir-injected or control circSRY KO mice testes (n=3 biologically independent experiments). Scale bar indicates 25 μ m.

(B) Representative image of immunofluorescence staining of MVH (red), γ H2AX (green) and DAPI (blue) in antagomir-injected or control circSRY KO mice testes (n=3 biologically independent experiments). Scale bar indicates 25 μ m.

(C) Quantitative analysis of MVH positive cells/SOX9 positive cells ratio between antagomir-injected and PBS-injected control testis. (ns, not significant; unpaired, Student's *t*-test; a total of 56 tubules from 3 biologically independent experiments were counted).

(D) Quantitative analysis of γ H2AX intensity between antagomir-injected and PBS-injected control testis. (*****P* <0.0001; unpaired, Student's *t*-test; a total of 1082 cells from 3 biologically independent experiments were counted).

Furthermore, it is unclear why only a subset of testes were chosen in Fig. EV5H (6 testes), when the authors had many samples from which to choose. If there was truly a significant increase in γ H2AX expression, a larger sample size would have strengthened this analysis; conversely, if they were to use all the samples and would find a large variability that results in a non-significant difference, then the current limited analysis appears to be misleading.

Response: Indeed, a larger sample size would offer stronger proof for an increased γ H2AX expression. However, due to insufficient experimental materials for performing all the related assays using one testes sample, we randomly selected 6 testes for γ H2AX expression detection, the remaining was used for paraffin embedding and staining experiment of MVH、SOX9 and γ H2AX.

Because the correlation between miR-138-5p and γ H2AX was not strongly demonstrated, we have changed our conclusion in the last paragraph into “In summary, our results suggest that circSRY acts as a sponge to competitively bind miR-138-5p *in vivo*, and the loss of circSRY leads to reduced γ H2AX level and defected meiosis, which further impairs spermatogenesis in adult mice.” We also changed the title with “Loss of circSRY reduces γ H2AX level in germ cells and impairs mouse spermatogenesis”.

Finally, are there any positive controls available to verify that the antagomir actually worked to reduce miR-138-5p activity? Overall, there are several major concerns with the antagomir assays.

Response: Antagomir is a modified inhibitor of miR-138-5p, of which activity on

reducing miR-138-5p has been shown *in vitro* (Fig. 6H). Furthermore, antagomir is a validated commercial product, which has been reported in related studies to inhibit miR-138-5p activity *in vivo* (Morton et al., 2008; Nama et al., 2019).

At a minimum, any claims about links between H2ax and miR-138-5p should be softened.

Response: We agree with your comments. Due to the weak correlation between miR-138-5p and γ H2AX, we have toned down our conclusion in the last paragraph into “In summary, our results suggest that circSRY acts as a sponge to competitively bind miR-138-5p *in vivo*, and the loss of circSRY leads to reduced γ H2AX level and defected meiosis, which further impairs spermatogenesis in adult mice.” We also changed the title with “Loss of circSRY reduces γ H2AX level in germ cells and impairs mouse spermatogenesis”.

2. In Figure 2F, the authors need to quantify cell death (TUNEL+ cells).

Response: Thank the reviewer for the helpful suggestion. We performed immunostaining of DDX4 and TUNEL assay to quantify the death cells within seminiferous tubules. The number of TUNEL-positive cells were significantly higher than control group. We also calculated the ratio of apoptotic cells to DDX4-positive cells. The ratio of TUNEL-positive cells to DDX4-positive cells was significantly higher than control group. The data were shown in Figure 2F.

DDX4 and TUNEL co-staining in circSRY KO and control mice testis

(A) Germ cells were labeled with the antibody against DDX4 (red) and demonstrated TUNEL assay (green) in circSRY KO and control 9-week-old mouse testes. Scale bar indicates 50 μm . (B and C) Number of TUNEL-positive cells or ratio of TUNEL-positive cells to DDX4-positive cells within seminiferous tubules in circSRY KO and control 9-week-old mouse testes. The total tubule number reached 100 from more than 3 biologically independent experiments ($****P < 0.0001$; unpaired, Student's *t*-test).

3. In Fig. 2I, 3B, and EV3B (and elsewhere), the y-axis label units are misleading for the sperm counts since it jumps from 0 to 5×10^6 . This scale is somewhat misleading in terms of the magnitude of change.

Response: We thank the reviewer for pointing out this issue. We have changed the scale of Y axis of Figure 2I, 3B, 3F and Figure S2B(EV2B), S3D(EV3D), S5F(EV5F) to start from 0 to avoid misleading display.

4. Was it surprising in Fig. 4C that there was no reduction in haploid (1N) cells, since there was a reduction in 4N cells and a reduction in sperm count in the CircSry cKO

mice? These findings don't quite add up.

Response: The values shown in Fig.4C indicate the relative percentage within the group, thus a reduction in 4N would lead to a relative increase in 1N or 2N cells. A reduction in 4N percentage only suggests a proportional change. Actually, the numbers of 1N, 2N and 4N cells counts in circSRY KO group all decreased evidenced by a longer time to collect the same total cells in each group for flow cytometry analysis (data not shown). For a clear demonstration, we added mean \pm SEM from 3 biologically independent experiments. The revised figure is shown below.

Fig.4(C) Flow cytometry analysis of the proportion of 4N cells, 1N cells, and 2N cells in 9 weeks mice testes between control and circSRY KO mice (* $P < 0.05$; unpaired, Student's t -test; $n=3$ biologically independent experiments).

5. Are the gene expression changes in Fig. 5G statistically significant?

Response: We thank the reviewer for pointing out this issue. Because we only obtained a small sample size ($n=2$), statistical analysis is not suggested. We thus showed the individual values in the figure below and circSRY KO spermatocytes showed higher mean values for all three genes than control group. We then did qRT-PCR analysis and verified the results.

(left) TPM values of *Rbbp7*, *Cul4b*, *Eif2s3y* from RNA-seq (n=2 mice), and (right) Validation of expression of X-linked genes, *Rbbp7*, *Cul4b*; Y-linked genes, *Eif2s3y* from control versus 9-week-old circSRY KO spermatocytes (* $P < 0.05$; ns, not significant; unpaired, Student's *t*-test; n=3 biologically independent experiments).

6. Statistics should be performed and significant differences should be denoted in Fig. 6B.

Response: We thank the reviewer for pointing out this issue. We have performed the statistical analysis in Figure 6B. The data were shown below.

Luciferase reporter assay

Luciferase reporter assay showed that 6 miRNAs were capable of binding circSRY and reduced the luciferase signal of luci-circSRY (**** $P < 0.0001$, *** $P < 0.001$, ** $P < 0.001$, unpaired, Student *t*-test, n=4 biologically independent experiments).

7. Detailed information on mouse strains used should be included in the Methods section.

Response: Thank the reviewer for the suggestion. We have added information on mouse strains used in our manuscript, please see the Materials and Methods section.

8. Minor point: GAPDH should be labeled as Gapdh in qRT-PCR graphs, since it refers to RNA and not protein.

Response: Thank the reviewer for pointing out this error. We have replaced GAPDH with Gapdh in Figure 2C and Figure 5H.

9. Minor point: There are still many remaining minor typographical and grammatical errors in the manuscript. These likely can be addressed during copyediting.

Response: Thank the reviewer for the helpful suggestion. We have carefully revised our manuscript and corrected typographical and grammatical errors.

Reference

Morton, S.U., Scherz, P.J., Cordes, K.R., Ivey, K.N., Stainier, D.Y., and Srivastava, D. (2008). microRNA-138 modulates cardiac patterning during embryonic development. *Proceedings of the National Academy of Sciences of the United States of America* *105*, 17830-17835. 10.1073/pnas.0804673105.

Nama, S., Muhuri, M., Di Pascale, F., Quah, S., Aswad, L., Fullwood, M., and Sampath, P. (2019). MicroRNA-138 is a Prognostic Biomarker for Triple-Negative Breast Cancer and Promotes Tumorigenesis via TUSC2 repression. *Scientific reports* *9*, 12718. 10.1038/s41598-019-49155-4.

October 26, 2022

RE: Life Science Alliance Manuscript #LSA-2022-01617-TR

Prof. Haoyi Wang
Chinese Academy of Sciences

Dear Dr. Wang,

Thank you for submitting your revised manuscript entitled "Loss of circSRY reduces γ H2AX level in germ cells and impairs mouse spermatogenesis". We would be happy to publish your paper in Life Science Alliance pending final revisions necessary to meet our formatting guidelines.

- please add ORCID ID for both corresponding authors-you should have received instructions on how to do so
- please add the Twitter handle of your host institute/organization as well as your own or/and one of the authors in our system
- please add the author contributions to the main manuscript text
- please add your supplementary figure legends and table legends to the main manuscript text
- please make sure your table files are uploaded as editable doc or excel files; it is fine if the tables remain in the uploaded 'Supplemental Material' doc file

Figure Check:

- please add sizes next to all blots

A. FINAL FILES:

B. MANUSCRIPT ORGANIZATION AND FORMATTING:

**Submission of a paper that does not conform to Life Science Alliance guidelines will delay the acceptance of your

manuscript.**

The license to publish form must be signed before your manuscript can be sent to production. A link to the electronic license to publish form will be sent to the corresponding author only. Please take a moment to check your funder requirements.

Sincerely,

November 7, 2022

RE: Life Science Alliance Manuscript #LSA-2022-01617-TRR

Prof. Haoyi Wang
Chinese Academy of Sciences
600 Main Street
Bar Harbor, ME 4609
China

Dear Dr. Wang,

Thank you for submitting your Research Article entitled "Loss of circSRY reduces γ H2AX level in germ cells and impairs mouse spermatogenesis". It is a pleasure to let you know that your manuscript is now accepted for publication in Life Science Alliance. Congratulations on this interesting work.

DISTRIBUTION OF MATERIALS:

Again, congratulations on a very nice paper. I hope you found the review process to be constructive and are pleased with how the manuscript was handled editorially. We look forward to future exciting submissions from your lab.

Sincerely,
